# Lower tropospheric distributions of $O_3$ and aerosol over Raoyang, a rural site in the North China Plain

Rui Wang[1], Xiaobin Xu[1], Shihui Jia[1,*], Ruisheng Ma[2], Liang Ran[3], Zhaoze Deng[3], Weili Lin[4], Ying Wang[1], Zhiqiang Ma[5]

[1] State Key Laboratory of Severe Weather & Key Laboratory for Atmospheric Chemistry of China Meteorological Administration, Chinese Academy of Meteorological Sciences, Beijing, China

[2] Guangxi Meteorological Disaster Mitigation Institute, Nanning, Guangxi, China

[3] Key Laboratory of Middle Atmosphere and Global Environment Observation, Institute of Atmospheric Physics, Chinese Academy of Sciences, Beijing, China

[4] Meteorological Observation Center, China Meteorological Administration, Beijing, China

[5] Institute of Urban Meteorology, China Meteorological Administration, Beijing, China

*now at School of Environment and Energy, South China University of Technology, Guangzhou, Guangdong, China

*Correspondence to*: Xiaobin Xu (xuxb@camscma.cn)

**Abstract.** The North China Plain (NCP) has become one of the most polluted regions in China, with the rapid increasing economic growth in the past decades. High concentrations of ambient $O_3$ and aerosol have been observed at urban as well as rural sites in the NCP. Most of the in situ observations of air pollutants have been conducted near the ground so that current knowledge about the vertical distributions of tropospheric $O_3$ and aerosol over the NCP region is still limited. In this study, vertical profiles of $O_3$ and size-resolved aerosol concentrations below 2.5 km were measured in summer 2014 over a rural site in the NCP using an unmanned aerial vehicle (UAV) equipped with miniature analyzers. In addition, vertical profiles of aerosol scattering property in the lower troposphere and vertical profiles of $O_3$ below 1 km were also observed at the site using a LIDAR and tethered balloon, respectively. The depths of the mixed layer and residual layer were determined according the vertical gradients of LIDAR particle extinction and aerosol number concentration. Average $O_3$ and size-resolved aerosol number concentration in both the mixed and residual layer were obtained from the data observed in seven UAV flights. The results show that during most of the flights the $O_3$ levels above the top of mixed layer were higher than those below. Such positive gradient in vertical distribution of $O_3$ makes the residual layer an important source of $O_3$ in the mixed layer, particularly during morning when the top of mixed layer is rapidly elevated. In contrast to $O_3$, aerosol number concentration was normally higher in the mixed layer than in the residual layer, particularly in early morning. Aerosol particles were overwhelmingly distributed in the size range <1 μm, showing slight differences between the mixed and residual layers. Our measurements confirm that in the lower troposphere over the rural area of the NCP is largely impacted by anthropogenic pollutants locally emitted or transported from urban areas. Compared with the historic $O_3$ vertical profiles over Beijing from the Measurement of Ozone and Water Vapor by Airbus In-Service Aircraft (MOZAIC), a strong increase in $O_3$ can be found at

all heights below 2.5 km in the decade from 2004 to 2014, with a largest enhancement of about 41.6 ppb. This indicates that the lower troposphere over the north part of the NCP has experienced rapid worsening photochemical pollution. This worsening trend in photochemical pollution deserves more attention in the future.

## 1 Introduction

Ozone ($O_3$) is a key trace gas and oxidant in the troposphere, which can generate hydroxyl radical (OH) that affects the oxidizing capacity of the atmosphere. Ground-level $O_3$ causes deleterious impacts on human health and vegetation (Anenberg et al., 2009; Mckee, 1994; Yue et al., 2014; Feng et al., 2008). Meanwhile, tropospheric $O_3$ is an effective greenhouse gas. The fifth IPCC assessment report points out that the increase of $O_3$ in the troposphere has caused + 0.40 (±0.20) W m$^{-2}$ of radiative forcing (Myhre et al., 2013). The vertical distribution of $O_3$ can influence the accumulation of $O_3$ in the surface layer and chemical reactions as well as radiation budget at different altitudes. The vertical distribution of $O_3$ is influenced by chemical and meteorological processes and varies with time and location (Kleinman et al., 1994; Fast et al., 1996; Lin et al., 2007; Ma et al., 2011). Therefore, direct measurements are needed to acquire the knowledge about the vertical distribution of O3, which is important to understanding atmospheric chemistry and $O_3$ radiative forcing.

Atmospheric aerosols influence the climate by direct effect and indirect effect (Schwartz, 1996; Coakley et al., 1983; Kiehl and Briegleb, 1993; Charlson, et al., 1997; Tegen et al., 2000). High loading of aerosols causes poor visibility and air quality. The diameters of aerosol particles span over four orders of magnitude, from a few nanometers to around 100 μm (Seinfeld and Pandis, 2006). Aerosol size distribution is one of the most critical factors determining light scattering property, health effects, and lifetime of aerosols. Observations of aerosol size distribution and its spatiotemporal variations have been one of the important aspects in the study of atmospheric aerosols.

The planetary boundary layer (PBL) is the lowest part of the troposphere, which is influenced to a large extent by the friction of earth's surface and objects on it (Ahrens, 2011). The PBL is directly impacted by most of anthropogenic and natural sources of trace gases and aerosols. In the PBL, the air turbulence can be so strong that it drives the rapid exchange of heat, water vapor, trace gases and aerosols between the atmosphere and earth's surface. The structure of the PBL is critical to the vertical

diffusion and transport of air pollutants so that the changes of vertical distributions of many trace gases (such as $O_3$) and aerosols are closely related to the evolution of the PBL.

The diurnal evolution of the PBL usually leads to a well-developed mixed layer in the daytime (He and Mao, 2005) and a stable layer overlaid by a residual layer in the nighttime. The formation of nocturnal stable layer may exert significant impacts on the concentrations of species in the surface layer. For example, surface $O_3$ is substantially removed during night by reactions with NO and some other species, and by dry deposition, particularly in urban areas during cold seasons (Lin et al., 2011). Following the development of the mixed layer on the next day, $O_3$ in the residual layer can be downward mixed, making a contribution to the enhancement of ground-level $O_3$ and even driving the photochemical processes (Aneja et al., 2000; Kleinman et al., 1994; Neu et al., 1994; Zhang et al., 1999). Therefore, the $O_3$ distribution in both mixed layer and residual layer deserves attention. Similarly, chemical compositions and physical properties including size distribution of aerosols may also be impacted by the evolution of PBL. Knowledge of this aspect is important for atmospheric chemistry and physics studies.

In recent decades, with the rapid economic development and the urbanization in China, ground-level and tropospheric $O_3$ and aerosol have increased significantly, especially in the North China Plain (NCP) (Ding et al., 2008; Xu and Lin, 2011; Ma et al., 2016; Chen et al., 2015; Ding and Liu, 2014). So far, observational studies on $O_3$ and aerosol in the NCP have been mainly conducted in the surface layer. There is only limited knowledge of the vertical distributions of tropospheric $O_3$ and aerosol over the NCP region, gained in some sporadic observational studies (Chen et al., 2009; Chen et al., 2013; Ma et al., 2011; Wang et al., 2012; Zheng et al., 2005). This hinders extensive validations of atmospheric chemistry models as well as the assessment of climate effects of $O_3$ and aerosols over this important region.

The detection of vertical distributions of $O_3$ and aerosol can be made using balloon sounding, aircraft, and tethered balloon. Each of the techniques has its advantages and disadvantages. Balloon sounding is a good way of obtaining $O_3$ and aerosol vertical profiles below about 30 km, but the sondes for $O_3$ and aerosol are usually expensive and not reusable, not to mention ground facilities needed for the sounding. Tethered balloon attached with $O_3$ and aerosol devices can be used for many times of observations under stable weather conditions, but it is usually not suitable to be operated above 1 km and needs manpower to run it carefully. Aircraft equipped with analyzers for $O_3$, other trace gases and aerosol instruments can be used to

simultaneously obtain distributions of these species along the flight tracks, but such observation is normally expensive and needs strong logistic supports of airport.

Typically, most aircraft measurements are made using manned aircrafts. However, unmanned aerial vehicles (UAVs) can also be used as research platforms. In the last decade, UAVs have been successfully used in the fields of atmosphere science and environment monitoring in order to understand the three-dimensional distribution of atmospheric species (Altstädter, et al., 2015; Illingworth et al., 2014; Watai et al., 2006; Rauneker and Lischeid, 2012; Gao et al., 2016). In comparison with manned aircrafts, UAVs have a few advantages, such as no need for airport, lower costs, higher flexibility, the possibility to investigate atmospheric parameters at small scales and low altitudes and the potential for application in regions dangerous for manned aircraft (Altstädter, et al., 2015). Above all, measurements with high spatial resolutions can be obtained using UAVs. For example, in a city center with a diameter of approximately 2 km, a typical instrument with 1 Hz resolution would only be able to make approximately 20 measurements during an overpass of the city when a normal atmospheric research aircraft was used. In contrast, more measurements would be obtained by an UAV so that the measurement uncertainty can be effectively reduced.

In this paper, we present UAV measurements of vertical distributions of $O_3$ and size-resolved aerosol number concentrations obtained over a site in the NCP during a field campaign. Data of surface $O_3$ and LIDAR observations of aerosol extinction from a nearby site are included in the analysis to facilitate interpretations and discussions of the measurements. To the best of our knowledge, there has been no similar published study from China.

## 2 Experimental

### 2.1 Ground-based measurements

Ground-based measurements, including measurements of surface $O_3$ and some other reactive gases ($NO/NO_2/NO_x$, $NO_y$, HCHO, PAN, $SO_2$, CO, $NH_3$, etc.), tethered balloon measurements of $O_3$ and black carbon (BC) vertical profiles, LIDAR observation of aerosol vertical profiles, etc., were conducted from 7 June to 18 August 2014 at the Raoyang Meteorological Station (RMS, 115°44′N, 38°14′N, 20m, a semirural site) in the NCP. RMS is located in Raoyang County, an agriculture county in the middle of Heibei Province. There are no large industrial sources in Raoyang. However, some influences on the

RMS site from the residence and traffic sources in the surrounding areas cannot be excluded. More details about the RMS site are given in Ran et al. (2016).

Surface $O_3$ was measured using a photometric $O_3$ analyzer (TE 49C, Thermo Electron, USA). The detection limit and precision of the $O_3$ analyzer are 1 ppb and 1 ppb, respectively. The analyzer was calibrated on 17 June, 16 July and 18 August, 2014 and showed no significant drifts. Details about the calibration and maintenance are described in Lin et al. (2009). A dual beam ozone monitor (Model-205, 2B Technologies, USA) was used for the tethered balloon observations of $O_3$ vertical profiles. The ozone monitor has dimensions of $9 \times 21 \times 29$ cm and weighs only 2.3 kg. It works based on the UV absorption at 254 nm at a maximum frequency of 0.5 Hz. It measures $O_3$ with a precision of 1 ppb or 2% of reading and has a limit of detection of 1 ppb. The ozone monitor was calibrated at the site using an $O_3$ calibrator (TE 49i-PS, Thermo Electron, USA).

Vertical profiles of aerosol were observed using a LIDAR (Leosphere, France). The LIDAR principle is based on the scattering phenomenon of light. A laser pulse is sent into the atmosphere and scattered by the target molecules or particles (clouds, dust, soot particles etc.). The backscattered light is collected by an optical system and its intensity is measured by a photo-detector, converted into an electronics signal and stored onto a computer. The wavelength of the laser sent is 355 nm. The vertical resolution of the LIDAR measurements is 15m and data below 200 m are discarded (EZ Aerosol and cloud LIDAR user manual).

## 2.2 Flight information

An UAV platform was used for the vertical profile measurements. The body of the UAV is made of glass fiber reinforced plastics, with a wingspan of 3.2 m. The UAV is battery-powered so that there is no contaminant from the engine exhaust of the UAV. The cruising speed and maximum cruising altitude of the UAV are 25 m/s and 5.5 km, respectively. As we were more interested in the vertical distributions of $O_3$ and aerosol in the lower troposphere, the UAV was programmed to fly below 3 km over an area within 5 km in diameter.

A miniature ozone monitor, personal ozone monitor (POM, 2B Technologies, USA) and handheld optical particle counter (OPC) (Handheld 2016 or 3016, Lighthouse, USA) were installed in the small cabinet of the UAV to measure the $O_3$ and aerosol number concentrations on the flight. The POM has dimensions of 10.2 x 7.6 x 3.8 cm and weighs only 0.34 kg. It works

based on the UV absorption at 254 nm at a maximum frequency of 0.1 Hz. It measures $O_3$ with a precision of 2 ppb or 2% and has a limit of detection of 4 ppb, well below the normal $O_3$ level over the Raoyang site. The POM was calibrated in the factory against a NIST-traceable standard instrument and compared at the site with the TE 49i-PS $O_3$ calibrator. The handheld OPC has dimensions of 22.23 x 12.7 x 6.35 cm and weighs about 1 kg. The Handheld 2016 counts aerosol numbers in the size-bins

0.2-0.3μm, 0.3-0.5μm, 0.5-0.7μm, 0.7 -1.0μm, 1.0-2.0μm and >2μm, while the Handheld 3016 (used only in the flight during 5:53-6:18 of July 31) records aerosol numbers in the size-bins 0.3-0.5μm, 0.5-1.0μm, 1.0-3.0μm, 3.0-5.0μm, 5.0-10.0μm and >10μm in differential or accumulation method. The OPC devices were set to collect data at 1 Hz. An isokinetic tube was mounted on the nose of the UAV to introduce ambient air into the OPC. In addition, a temperature/humidity sensor was attached to the OPC so that the sampling flow rate was converted to standard condition. Both OPC devices were calibrated

against a NIST-traceable source. More details are given in the supplement (section S1).

Limited mainly by the availability of airspace, only seven successful UAV measurements were conducted several kilometers west of the RMS during the observation period. More detailed flight information is summarized in Table 1.

## 3 Results and discussion

### 3.1 Vertical profiles of $O_3$, aerosol number concentration and extinction

The vertical profiles of $O_3$ were acquired during seven UAV flights as listed in Table 1, while aerosol number concentrations in six size-bins were observed during five of the seven flights (Flights 1-5). The observed aerosols particles were predominately distributed in the accumulation mode since it is not possible to count the aerosol numbers in the Atkin mode using the handheld OPC. Figure 1 shows the vertical profiles of $O_3$ and aerosol number concentration observed during all flights. To study how the evolution of PBL influences $O_3$ and aerosol concentration at different altitudes, the vertical profiles in Fig.1 are grouped in

early morning (5:40-8:00 LT), late morning (10:00-12:00 LT) and afternoon (15:00-19:00 LT) periods and shown in Figs. 1(a), 1(b) and 1(c), respectively. In addition, average particle extinction vertical profiles simultaneously obtained by the LIDAR are presented, with data points less than or equal to zero being excluded. All the vertical profiles shown in Fig. 1 include averages over 50 m vertical layers, regardless of the spatial resolutions of actual observations. Although the site over which the flights

were conducted is several kilometers away from the RMS, both sites were influenced by the same synoptic system. Therefore, the results (Fig.1) from both sites should be comparable within the observation uncertainties. In the early morning of July 29, 2014, particle extinction of more than 3000 $Mm^{-1}$ was observed at around 1600 m (Fig. 1(a3)), indicating the presence of cloud there (Huang et al., 2011).

It is noteworthy in Fig. 1 that aerosol number concentrations during late morning of June 29 (Flights 1) and early morning of July 1 (Flight 3) and the $O_3$ mixing ratio during late morning of June 29 (Flight 1) were significantly lower than those during other flights, and their vertical profiles were slightly different from others. This indicates that some factors might have impacted the levels and vertical profiles of $O_3$ and aerosol. To understand those phenomena, we display the airflow fields at 1000 hPa and 850 hPa over the area surrounding RMS in Fig. S2 in the supplement and 48-h backward trajectories

of air parcels arriving at 100 m, 500 m, 1000 m, 1500 m and 2000m over RMS in Fig. S3 in the supplement for 8:00 local time of June 29, July 1, July 29 and July 31, 2014, calculated using the Hybrid Single-Particle Lagrangian Integrated Trajectory (HYSPLIT) model (Draxler and Rolph, 2003).

Figures S2(a) and S2(b) indicate that the 1000 hPa and 850 hPa levels on early morning of June 29 were dominated by different air circulations. Figure S3(a) shows that the air parcels arriving at 100 m and 500 m over RMS were from the

boundary layer over the nearby areas and mainly from south of the site, while those arriving at 1000 m and above originated far from the north (northern Heilongjiang) and travelled rapidly over 2000 m for most of the time. The disparate airflows in the bottom and upper layers made a large negative gradient in the vertical distributions of $O_3$ and aerosol number concentration during Flight 1, as can be seen in the vertical profiles shown in Figs. 1(b1) and 1(b2).

In the whole layer, the aerosol number concentration in the early morning of July 1 (Flight 3) was much lower than those in

the other two flights (Flight 4 and 5) in the early morning of July 29 and July 31 (Fig. 1(a2)). Figures S2(c) and S2(d) in the supplement show that the airflow at the 1000 hPa and 850 hPa levels over RMS was mostly from the south-southwest. Figure S3(b) in the supplement indicates that air parcels arriving at different heights over RMS originated either from south or from west or east bending to south. Using the Meteorological Information Comprehensive Analysis and Process System (MICAPS, http://www.cma.gov.cn/en2014/20150311/20160615/index.html) we found that there was rainfall within the 48 hours prior to

5:00 LT of July 1 over the north, south by southwest and the southeast of RMS. It can be inferred that in the early morning of

July 1 air transported to the lower troposphere over RMS had been mixed with cleaner air during the rainy conditions so that the aerosol number concentration declined substantially. Therefore, synoptic situations are important factors influencing the concentrations of air pollutants and their vertical distributions. Another important factor is the PBL evolution, as discussed in the next section.

## 3.2 Determination of the mixed and residual layer depth

The PBL development can, in principle, be determined by measurement of vertical profiles of a variety of atmospheric properties, with potential temperature being the most common one (Lenschow, 1986). Vertical aerosol number concentration vertical profiles allow the determination of the PBL, as aerosol acts as a tracer of atmospheric turbulent forces along height (Ferrero et al., 2011, 2014). One powerful way to probe the PBL is based on vertical profiles of particle extinction detected using LIDAR. There are several methods to explore the structure of PBL from vertical profiles of particle extinction, such as the calculation of vertical gradient of particle extinction (He and Mao, 2005; Huang et al, 2011; Hayden et al., 1994; Hoff et al., 1996; Melfi et al., 1985), wavelet analysis (Cohn et al., 1997; Brooks et al., 2003), and parameterization (Steyn et al., 1999). In this study, calculations of the vertical gradient of LIDAR particle extinction ($dE/dz$) and particle number concentration ($dN/dz$) are adopted to identify the mixed layer depth (MLD) and residual layer depth (RLD).

To obtain a general picture of the diurnal PBL evolution, particle extinction values at different heights in the lower troposphere (0-3 km) were averaged for different time intervals on all sunny days during the campaign. The average vertical profiles in a half hour and two hour intervals are displayed in Figs. 2 and 3, respectively. The zero and negative values in the LIDAR measurements were excluded, which may lead to averages slightly overestimated. Averages with less than 75% data availability were rejected to avoid the impact of under-sampling on the average vertical profiles. As mentioned in section 1, the PBL experiences a diurnal cycle. The mixed layer develops after the sunrise. Its top rises continually up and may exceed the top of the residual layer formed on the previous day. After the sunset, the mixed layer collapses gradually, forming the nocturnal stable layer and residual layer.

Figures 2 and 3 clearly show the average PBL diurnal cycle of vertical distribution of particle extinction. According to the solar elevation angles, the sunrise and sunset time is estimated to be 5:00-6:00 LT and 19:00-20:00 LT, respectively. In Fig. 3,

the maximum of particle extinction was 0.53 km-1 at 660 m after the sunset and the top of residual layer was about 705 m. Moreover, the nocturnal stable layer started forming, reaching a depth of about 300 m at 0:00-1:00 LT on the next day (Fig. 3(a)). During the night, the top of average residual layer fluctuated slightly around 700 m. After sunrise on the next day, the nocturnal stable layer disappeared gradually, and the mixed layer developed. The top of mixed layer reached about 840 m at around 11:00 LT and stayed until 13:00 LT, when it started declining. The particle extinction in the mixed layer increased during the morning period, presumably because of the increasing aerosol emissions, downward mixing from the residual layer, and photochemical production from precursor gases.

The data shown in Figs. 2 and 3 are consistent with our understanding of the diurnal PBL evolution, suggesting that LIDAR data captured the structure of PBL and can be used in the determination of the MLD and RLD. Based on individual dE/dz vertical profiles of particle extinction, we can determine the tops of the mixed layer and residual layer for the periods with UAV flights by identifying the minima below 2000 m in the dE/dz vertical profiles if there were no obviously unreasonable values in the vertical profiles due to interferences or noise. In order to select the minima that were really resulted from a strong decrease of aerosol burden for a small height change rather than abnormal variation, we compared the minima before and after the flights with those during the flights. Finally, the MLD and RLD were obtained according to the heights at which the minima were considered to be the most reasonable ones. However, this method was not applicable when the MLD was in the blind zone of LIDAR (lower than 200m) or there was cloud near the top of residual layer.

The vertical profile of particle extinction and its gradient for the flight (10:18-10:47 LT) in the late morning of June 29 (Flight 1) are displayed in Fig. 4 to show how to determine the MLD and RLD. To facilitate the identification of the mixed layer and residual layer, the vertical profiles of particle extinction before and after the flight are also shown in Fig. 4(a). With the consideration of turbulent time scale in the atmosphere, the particle extinction vertical profiles at 8:00 LT and 12:00 LT were calculated from the LIDAR observations in the periods 7:30-8:30 LT and 11:30-12:30 LT, respectively. As can be seen in Fig. 4(b), there are a few minima in the calculated gradient, two of which (marked with filled black circles) may represent the tops of the mixed layer and residual layer. Comparing the vertical profile of particle extinction during the flight with those before and after the flight, we can see gradual changes in heights of the inflection points in the lower parts of the vertical

profiles (0-1500 m) with the evolution of the PBL. This makes us more certain that the filled black circles in the gradient curve in Fig. 4(b) indicate the tops of the mixed layer and residual layer.

Similar to the vertical gradient of particle extinction, the vertical gradient of aerosol number concentration can also be used to probe the PBL. To find out the key inflection points in the vertical profiles, which indicate the tops of the mixed layer and residual layer, we paid attention to the major structures in vertical profiles and gradients of aerosol number concentration during the ascent and descent flights. Due to the slight differences in time and space in the flight, there were some differences in particle number concentrations measured at the same height between the ascent and descent of a flight. Even though, the t-test ($\alpha$=0.05) may help to judge whether or not the results are accordant. As an example, Fig. 5 shows the aerosol number concentration data from a flight in the early morning of July 29 (Flight 4). The two minima at 275 m and 1770 m (marked with red solid circles) in the gradient curve (Fig. 5(b)) corresponds obviously the major inflection points in the vertical profile of ascent flight (Fig. 5(a)). In the gradient curve from the descent flight, there are also two minima below 2000 m. These two minima do not overlay those from the ascent flight, but there are no significant ($\alpha$=0.05) differences in the minimum positions between ascent and descent. Therefore, the marked heights 275 m and 1770 m are considered to be the tops of mixed layer and residual layer, respectively.

Determining the MLD and RLD facilitates the calculations of average $O_3$ and aerosol number concentrations in the mixed layer and the residual layer. Table 2 summarizes the mixed and residual layer heights, determined using the methods discussed above, and the calculated $O_3$ and aerosol number concentrations for the two layers during the flights. The MLD or RLD during some flights cannot be determined in the case that the MLD was in the blind zone of LIDAR (lower than 200m) or there was cloud near the top of residual layer. Ideally, the mixed and residual layer heights determined using the two methods were the same, so would be the average $O_3$ and aerosol number concentrations. However, the real conditions were not idealized. As can be seen in Table 2, the two methods produced different results of the MLD and RLD, and the average $O_3$ and aerosol number concentrations.

To see how the results from the two methods are different from each other, we conducted correlation analyses. In Fig. 6, average $O_3$ and aerosol number concentrations obtained using the method of particle extinction gradient (Method 1) are compared with those obtained using the method of particle number concentration gradient (Method 2). As can be seen in the

figure, the averages obtained using the two methods are highly significantly correlated. Linear regressions suggest that the slopes of correlation lines for average $O_3$ and aerosol number concentrations in the mixed layer and residual layer are all close to 1.0. In other words, the two methods used for the determination of the MLD and RLD are reliable and can produce comparable results.

**3.3 Average $O_3$ and aerosol number concentrations in the mixed and residual layers**

Data in Table 2 indicate that the mixed layer height increased from early morning to late morning and to afternoon. The top of residual layer did not show such tendency. The mixing ratio of $O_3$ in the surface layer and mixed layer was lowest in the early morning and highest in the afternoon. This can be readily explained by the changes in photochemical production, NO titration, and dry deposition of $O_3$ in the course of the day. Photochemical reactions often produce much more $O_3$ around noon than in the other period, while NO titration and dry deposition remove $O_3$ more effective during the night and early morning, when the NO level is higher and the mixed layer is shallower (see Fig. S4 in the supplement). In most cases, the diurnal cycle of surface $O_3$ can be well explained by these factors, if advection is negligible. However, there may be additional factors that contribute significantly to the diurnal variations of $O_3$ in the mixed layers, at least in our cases. Table 2 shows that the level of $O_3$ in the surface and mixed layers were lower than that in the residual layer during all early morning flights (Flights 3-5). Data from the flight in late morning of June 29 (Flight 1) indicate that the $O_3$ level in the residual layer was slightly lower than those in the surface and mixed layers. Fig. 1(c1) shows that $O_3$ vertical profiles were relatively straight and the $O_3$ levels in the surface and mixed layers were close to or slightly lower than those at the heights above the mixed layer. These suggest that during most of our flights the $O_3$ levels above the top of mixed layer were higher than those below. Similar vertical $O_3$ profiles were also observed over a site in Beijing by Ma et al. (2011).

The positive gradient in vertical distribution of $O_3$ makes the residual layer an important source of $O_3$ in the mixed layer, particularly during morning when the top of mixed layer is rapidly elevated. Therefore, downward mixing of $O_3$ in the residual layer over our site may play an important role in the diurnal cycle of $O_3$ at ground-level and in the mixed layer. Previous studies show that $O_3$ in the free troposphere over the NCP can be rapidly downward transported to the surface layer, driven by high wind speed in winter (Lin et al., 2011; Zhang et al., 2014a) or nighttime convection processes in summer (Jia et al., 2015). Our

data show that even under normal meteorological conditions in summer, downward mixing of $O_3$ at higher altitudes over the NCP can be a significant contributor to the $O_3$ level in the mixed layer. The physical processes that influence surface $O_3$ are normally considered in modern air quality models. It is of interest to test how well the models can simulate the impacts of these processes on surface $O_3$. Our measurements provide valuable experimental data for validating model results and satellite retrievals, which is out of scope of this paper.

In contrast to the $O_3$ level, aerosol number concentration in early morning was higher in the mixed layer than in the residual layer. This can be attributed to the fact that the major sources of primary and secondary aerosols are in the mixed layer. With the PBL development during the daytime aerosol particles are mixed within a deeper layer as can be seen in Fig. 2.

### 3.4 Aerosol number size distributions in the mixed layer and residual layer

The mass concentration, size distribution, and chemical composition of aerosol are closely related with emissions, atmospheric chemistry, and meteorological conditions. The formation of the mixed layer and residual layer reduces the vertical mixing and may cause some differences in the physico-chemical properties of aerosols between both layers. Here, we investigate the difference in aerosol number-size distribution between the mixed and residual layers, taking the advantage of clear separation of both layers. Aerosol number concentrations in different size-bins were averaged for the mixed and the residual layers determined using aerosol number concentration gradients. Figure 7 shows the aerosol number and volume size distributions in the mixed layer and residual layer during early morning of July 1 (Flight 3), early morning of July 29 (Flight 4), late morning of June 29 (Flight 1) and afternoon of June 29 (Flight 2). The aerosol volume concentrations were calculated from the measured number size distributions under the assumption that all particles were spherical.

Obviously, the particle number concentration in the mixed layer was higher than that in the residual layer. This is expected as the mixed layer is normally more impacted by emission sources, which are mainly within the mixed layer. On the other hand, the differences of aerosol number and volume concentration between the mixed layer and the residual layer were more significant in the size range >2 μm in comparison with the other size ranges. This can be well explained by the fact that the mixed layer is much more influenced than residual layer by the ground-level emission of aerosol in coarse mode. Moreover, aerosol particles were overwhelmingly distributed in the size range <1 μm, independent of observation time. Such aerosol size distribution and ranges of aerosol number concentrations unsurprisingly resembled urban aerosol (Baron et al., 2011).

Although some coarse particles may be released by the farm over which our UAV observations were conducted, their contribution to our measurements of aerosol number concentration should be limited within the surface layer, considering the wind conditions during the UAV flights. In other word, the aerosol size distributions at higher altitudes of the PBL were nearly not influenced by local coarse particle emissions. In recent decades, the NCP region has been found to be one of the most polluted regions of the world, suffering severe haze pollution (Fu et al., 2014). Even in the rural areas of the NCP, air quality is largely impacted by anthropogenic pollutants locally emitted or transported from urban areas (Lin et al., 2008; Lin et al., 2009; Liu et al., 2010). Therefore, aerosol size distributions shown in Fig. 7 should represent well mixed aerosol over a rural area in the NCP, which is dominated by fine particles.

It is interesting to see that the patterns of aerosol number and volume size distributions in Flight 4 (early morning in July 29) are different from those in Flights 1 (late morning in June 29), 2 (afternoon in June 29) and 3 (early morning in July 1). The number size distributions in Flights 1, 2 and 3 show monotonic decreasing with increasing aerosol size (Figs. 7(a), 7(c) and 7(d)), while those in Flight 4 peak in the range of 0.3-0.5μm (Fig. 7(b)). The aerosol size distribution in the early morning of July 29 (Flight 4) was unique in that a larger portion of aerosol particles was found in the range 0.3-0.5μm. It may be caused by different synoptic conditions. Figure 1 indicates that among all the vertical profiles, the vertical profiles from July 1 (Flight 3) and July 29 (Flight 4) represent most clean and most polluted cases, respectively. Backward trajectories (Fig. S3 in the supplement) indicate that in early morning of July 29, air masses were from the southeast sector. Studies at other NCP sites also show that air masses from southeast are associated with higher particle number concentration (Shen et al., 2011) and contain aerosols of relatively larger mean medium diameter (Zhang et al., 2014c), which coincide with our results.

.

## 3.5 Comparison with historic $O_3$ vertical profiles

The increase in emissions of air pollutants in the NCP region has caused haze and $O_3$ pollution in recent decades. Emission control measures that have been introduced in this region have led to a decrease trend after 2003 in the annual number of haze days (Fu et al., 2014). However, the $O_3$ air quality has not been improved yet. In contrast, the maximum daily average 8-h (MDA8) mixing ratios of $O_3$ at a background site in the NCP increased at a rate of 1.13 ppb/yr from 2003 to 2015 (Ma et al.,

2016). The increase of $O_3$ level has taken place not only at ground-level, but also at other altitudes in the troposphere, particularly in the PBL. Ding et al. (2008) revealed a strong positive trend of the $O_3$ level in the boundary layer over Beijing from 1995 to 2005 using the Measurements of Ozone and Water Vapor by Airbus In-Service Aircraft (MOZAIC). Multi-year ozonesonde measurements over Beijing also indicate a significant increase of tropospheric $O_3$ during the period 2002-2010

(Wang et al., 2012).

It is of great interest to see how lower tropospheric $O_3$ has changed over the NCP region. Many locations in the NCP region are controlled by similar prevailing winds (southwesterly and northeasterly), particularly in summer (Lin et al., 2008; Lin et al., 2009). Although the MOZAIC measurements and our UAV observations of $O_3$ were conducted over two sites about 200 km apart from each other, $O_3$ vertical profiles in the lower troposphere over both sites may represent the situations over the north

part of the NCP. Therefore, it is assumed that the $O_3$ vertical profiles over both sites might be comparable. Under this assumption, we can try to obtain an idea about the change of summer $O_3$ in the lower troposphere over the NCP between the first MOZAIC measurements and our measurements.

Figure 8 displays average $O_3$ vertical profiles for different time period of the day in the lower troposphere obtained at RMS during our tethered balloon (Fig. 8(a)) and UAV observations (Fig. 8(b)) and summer daytime (5:00-19:00 LT) average $O_3$

vertical profiles from MOZAIC observations and our observations (Fig. 8(c)). The MOZAIC $O_3$ data have an estimated accuracy of $\pm[2\ ppb+2\%]$ for individual 4 s measurements and is suitable for building reliable $O_3$ climatologies (Thouret et al., 1998; Zbinden et al., 2013). MOZAIC $O_3$ vertical profile data for the greater Beijing area are available only for 1994-2005 with varying monthly vertical profile numbers and most of the vertical profiles were obtained during 5:00-18:00 LT period (Ding et al., 2008). To enhance the representativeness of average vertical profiles, all average MOZAIC $O_3$ vertical profiles were

calculated from at least 7 individual vertical profiles. For some years, no average MOZAIC $O_3$ vertical profiles were available due to inadequate MOZAIC measurements in summer. $O_3$ measurements from 52 tethered balloon experiments (TBE) over Raoyang were grouped and averaged for the early morning (5:00-10:00 LT), late morning (10:00-12:00 LT) and afternoon (12:00-19:00 LT) periods (Fig. 8(a)). Only 7 UAV experiments (UAVE) could be made during our campaign at Raoyang. Nevertheless, average $O_3$ vertical profiles were calculated for early morning, late morning and afternoon (Fig. 8(b)),

respectively. It is noted that only one UAV flight was made in late morning (Flight 1; 10:28-10:50 LT on June 29, 2014).

Therefore, the late morning average $O_3$ vertical profile in Fig. 8(b) is less representative. If this vertical profile is not considered, we can see in Figs. 8(a) and 8(b) a clear development of vertical $O_3$ distribution in the lower troposphere, with a substantial increase of the $O_3$ level in the mixed layer and a slight increase above the mixed layer in the course from the early morning to the afternoon. Although the mixing ratios of $O_3$ over Raoyang were mostly higher than 100 ppb in the afternoon, the overall average values were about 100 ppb as shown in the bottom-right plot of Fig. 8(c). Both our measurements and the MOZAIC measurements indicate high $O_3$ pollution in the lower troposphere over the NCP in summer.

Figure 8(c) shows that the $O_3$ level over the NCP had experienced a strong positive increase, indicating strengthening photochemical pollution in about two decades. The average mixing ratio of $O_3$ near the ground level had a relatively small increase (8.9 ppb) during 2004-2014, corresponding to an increase rate of about 0.9 ppb/yr. This increase rate is close to the average increase rate (1.1 ppb/yr ) reported by Tang et al. (2009) of surface $O_3$ at six urban/suburban sites in Beijing in July-September during 2001-2006, but only about one third of what Zhang et al. (2014b) found for August daytime surface $O_3$ at an urban site in Beijing during 2005-2011 (2.6 ppb/yr). However, the high-end value of $O_3$ near the ground level had a larger increase, as suggested by the right end of the error bars (Fig. 8(c)). This larger increase in the high-end value of $O_3$ is consistent with the large increase of the maximum daily average 8 h (MDA8) mixing ratio of $O_3$ at the Shangdianzi background station (Ma et al., 2016). Compared with the increase of $O_3$ near the ground-level, much larger increases were found in $O_3$ at higher altitudes in the lower troposphere over Raoyang during 2004-2014, with the maximum increase (41.6±15.5 ppb) being found at 1.5 km. Assuming that the $O_3$ level increased linearly over the ten years, the increase rate would be about 4.2±1.6 ppb/yr. Sun et al. (2016) compiled and analyzed the $O_3$, and $NO_x$ and CO data collected at the Mt. Tai site (36.25°N, 117.10°E; 1534m asl) during a few campaigns from 2003 to 2015. They reported that $O_3$ at Mt. Tai increased at 1.7±1.0 ppb/yr in June and 2.1±0.9 ppb/yr in July-August during 2003-2015. These rates of increase in summer $O_3$ at Mt. Tai and that we obtained for 1.5 km over Raoyang agree within the uncertainties though both sites are about 240 km apart. Based on the MOZAIC measurements over Beijing in summer afternoons (at 15:00–16:00 LT in MJJ) during 1995-2005, Ding et al. (2008) reported an increase rate of about 3 ppb/yr for $O_3$ in 0-2 km. From the data shown in Fig. 8(c), we can obtain an increase rate of 3.3 ppb/yr for summer $O_3$ in 0-2 km over Raoyang for the period 2004-2014, which agrees well with that reported by Ding et al. (2008). Note that the average $O_3$ vertical profile for summer 2014 (Fig. 8(c)) contains

measurements from the morning flights so that our estimated increase rate may be significantly lower than that for summer afternoon.

The above comparisons confirm that the abundance of $O_3$ in the lower troposphere over the north part of the NCP has largely increased since about two decades. The increase of the $O_3$ level in summer afternoon period seems to speed up after 2004. Network observations indicate that surface $O_3$ pollution in China's polluted regions, including the NCP, has become more severe in recent years in contrast with the apparent decreases of $PM_{2.5}$ and primary gaseous pollutants (http://www.cnemc.cn/publish/totalWebSite/0492/newList_1.html). Such trend in surface $O_3$ may exert significant impacts on human health and vegetation. The increase of $O_3$ in the lower troposphere may influence atmospheric chemistry, i.e., increase the oxidation capacity (Ma et al., 2012), and add radiation forcing over the region.

**4 Conclusion**

In this study, vertical profiles of particle extinction property, $O_3$ and size-resolved aerosol concentration were simultaneously measured using miniature devices installed in an UAV over a rural site in the NCP during 2014 summer, allowing the characterization of diurnal $O_3$ and aerosol concentration in mixing layer and residual layer over a small area. Seven vertical profiles were successfully obtained in this campaign.

We found a positive gradient in the vertical distribution of $O_3$, higher above the top of mixed layer than those below, making the residual layer an important source of $O_3$ in the mixed layer, particularly during morning when the top of mixed layer is rapidly elevated. However, aerosol number concentration in early morning was higher in the mixed layer than in the residual layer. The aerosol particles were abundant and overwhelmingly distributed in the size range <1 μm, consistent with urban aerosol. The historic $O_3$ data from the MOZAIC project, together with our measurements, were used to investigate the long-term changes in $O_3$ vertical profiles in the lower troposphere over the NCP. The comparison of data from the 2014 summer campaign with those from MOZAIC suggests that the $O_3$ level over the north part of the NCP has experienced a strong positive increase, and the increase of the $O_3$ level seems to speed up after 2004, indicating rapidly strengthening photochemical pollution particularly in the last decade. Observations show that surface $O_3$ pollution in China's polluted regions, including the

NCP, has become more severe in recent years in contrast with the apparent decreases of $PM_{2.5}$ and primary gaseous pollutants. In view of this, more attention should be paid to $O_3$ concentration under the control of PM.

## Data availability

The entire data set can be made available for scientific purposes upon request to the corresponding author.

*Acknowledgements.* This research was supported by the National Natural Science Foundation of China (No. 41330442) and China Special Fund for Meteorological Research in the Public Interest (No. GYHY201206015). We thank Zhiping Liu, Min Sun, Chaohui Wu for their assistance in the UAV observations and Yong Wang, Chuncheng Ji, Qihua Du, Hanze Yu, and Xinpan Li for their assistance in the tethered balloon observations. Logistic supports from the Raoyang Meteorological Bureau

are highly appreciated. The authors acknowledge the strong support of the European Commission, Airbus, and the Airlines (Lufthansa, Air-France, Austrian, Air Namibia, Cathay Pacific, Iberia and China Airlines so far) who carry the MOZAIC or IAGOS equipment and perform the maintenance since 1994. In its last 10 years of operation, MOZAIC has been funded by INSU-CNRS (France), Météo-France, Université Paul Sabatier (Toulouse, France) and Research Center Jülich (FZJ, Jülich, Germany). IAGOS has been additionally funded by the EU projects IAGOS-DS and IAGOS-ERI. The MOZAIC-IAGOS

database is supported by AERIS (CNES and INSU-CNRS). Data are also available via AERIS web site http://aeris-data.fr.

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

**Table 1.** The ascent and descent flight time (local time LT) and the max height.

| Flight | Date | Ascent period | Hmax(m) | Descent period |
|--------|------|---------------|-----------|----------------|
| 1 | June 29 | 10:18-10:36 | 1316 | 10:36-10:47 |
| 2 | June 29 | 17:58-18:24 | 2498 | 18:24-18:38 |
| 3 | July 1 | 6:34-6:55 | 2021 | 6:55-7:07 |
| 4 | July 29 | 7:35-7:51 | 2430 | 7:51-8:04 |
| 5 | July 31 | 5:53-6:18 | 2468 | 6:18-6:33 |
| 6 | July 31 | 17:26-17:44 | 2676 | 17:44-18:02 |
| 7 | August 2 | 15:17-15:29 | 2410 | 15:29-15:45 |

Table. 2. $O_3$ and aerosol number concentration calculated in method of particle extinction gradient (left of "/") and method of aerosol number concentration gradient (right of "/"). $O_3$ concentration on the surface is obtained in the surface measurement program.

| | | Early morning | | | Late morning | | Afternoon | |
|---|---|---|---|---|---|---|---|---|
| | | Jul.1 (Flight 3) | Jul.29 (Flight 4) | Jul.31 (Flight 5) | Jun.29 (Flight 1) | Jun.29 (Flight 2) | Jul.31 (Flight 6) | Aug.2 (Flight 7) |
| | | 6:34-7:07 | 7:35-8:04 | 5:53-6:33 | 10:18:10:47 | 17:58-18:38 | 17:26-18:02 | 15:17-15:45 |
| Surface | $O_3$(ppb) | 20.8±0.7 | 39.7±0.81 | 29.6±2.5 | 73.7±1.7 | 97.8±0.98 | 87.1±1.43 | 84.3±2.58 |
| Mixed layer | Height(m) | ::/200 | 255/275 | 255/267 | 360/281 | 795/525 | 795/:: | ::/765 |
| | $O_3$(ppb) | ::/37.4±20.8 | 44.3±8.5/44.3±8.5 | 70.5±38.6/72.4±38.2 | 71.2±9.0/69.1±9.6 | 106.7±7.9/103.2±9.2 | 99.1±5.4/:: | 104.2±4.9/:: |
| | N $(10^2 p/cm^3)$ | ::/1.99±0.36 | 8.7±0.12/8.69±0.13 | 5.73±2.00/5.73±2.00 | 4.17±0.75/4.54±0.75 | 6.28±0.17/6.39±0.10 | ::/:: | ::/:: |
| Residual layer | Height(m) | 1215/1331 | ::/1770 | 1110/1577 | 840/974 | ::/:: | ::/:: | ::/:: |
| | $O_3$(ppb) | 103.4±12.4/105.1±8.24 | ::/111±10.7 | 97±9.23/99.6±8.9 | 66.5±6.1/65.8±8.5 | ::/:: | ::/:: | ::/:: |
| | N $(10^2 p/cm^3)$ | 1.17±0.18/1.12±0.16 | ::/6.77±0.52 | 5.45±0.48/5.38±0.47 | 2.57±0.08/2.64±0.25 | ::/:: | ::/:: | ::/:: |

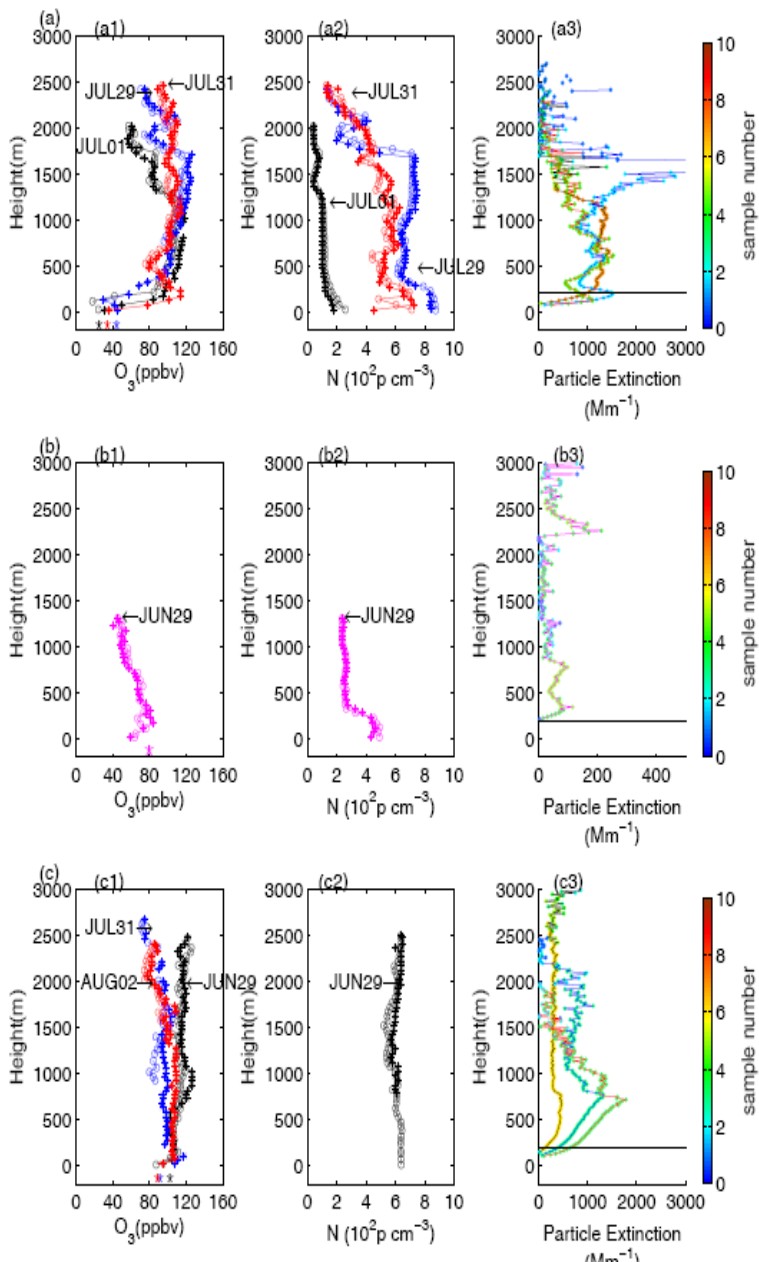

**Figure 1.** Vertical profiles of $O_3$ (a1,b1,c1), aerosol number concentration (a2,b2,c2) and particle extinction (a3,b3,c3) obtained in (a) early morning, (b) late morning, and (c) the afternoon, respectively. The $O_3$ and aerosol vertical profiles obtained during the ascent and descent of the UAV are indicated with circles and asterisks, respectively. Ground-level $O_3$ mixing ratios during the flights are also shown together with the $O_3$ vertical profiles (bigger asterisks with a negative offset in altitude). The color scales attached to particle extinction graphs (c1, c2, c3) show the sample numbers of individual particle extinctions, from which the averages of particle extinctions were calculated. The black lines indicate the altitude of 200 m above ground, under which the particle extinction data should not be used.

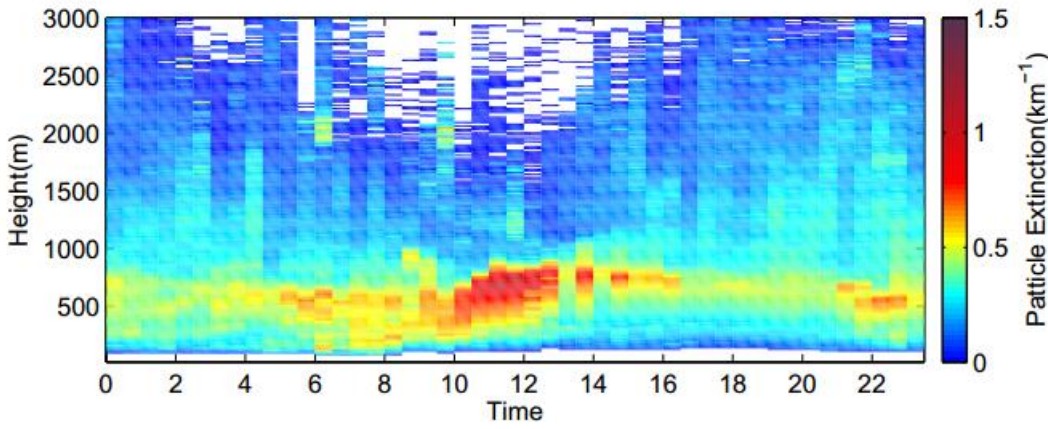

**Figure 2.** Average diurnal patterns of vertical distribution of particle extinction.

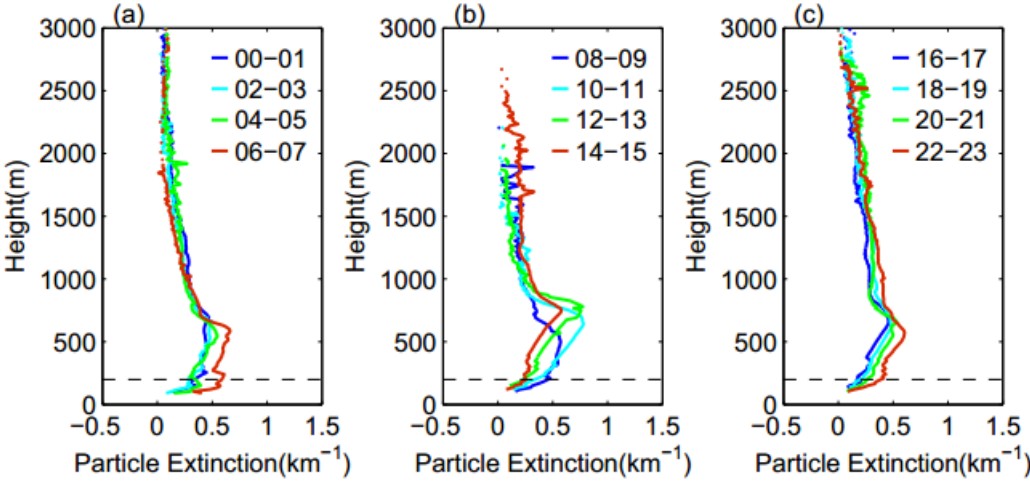

**Figure 3.** Vertical profiles of particle extinction averaged every two hours for (a) 0:00-7:00 LT, (b) 8:00-15:00 LT, and (c) 16:00-23:00 LT.

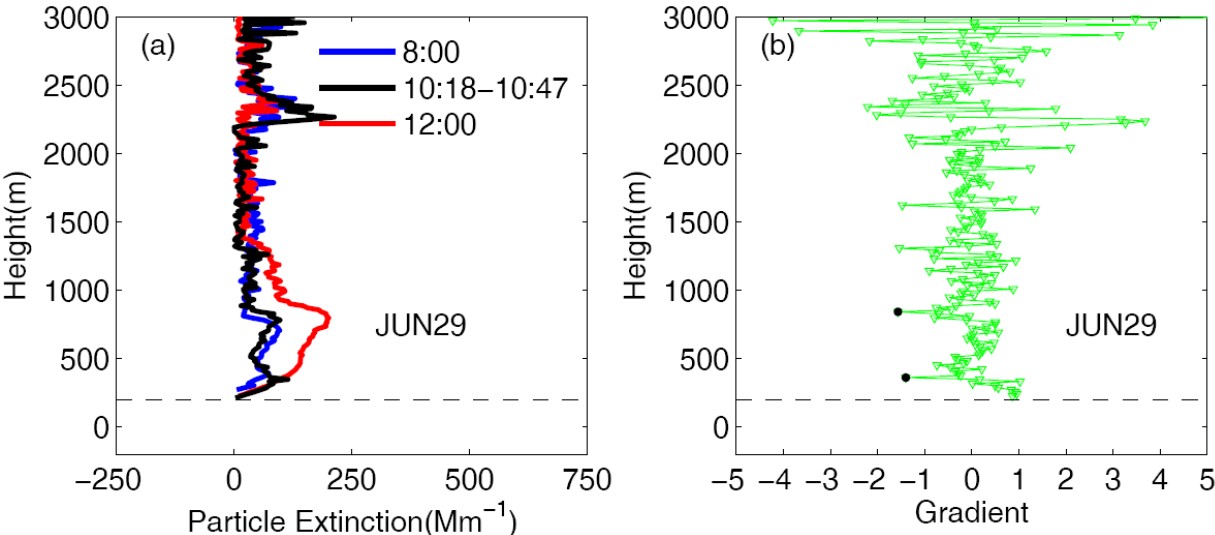

**Figure 4.** The vertical profile of particle extinction observed during the flight in the late morning of June 29 (a) and it gradient (b) and the vertical profiles of particle extinction before (blue) and after (red) the flight. The dashed lines indicate the lower limit of LIDAR.

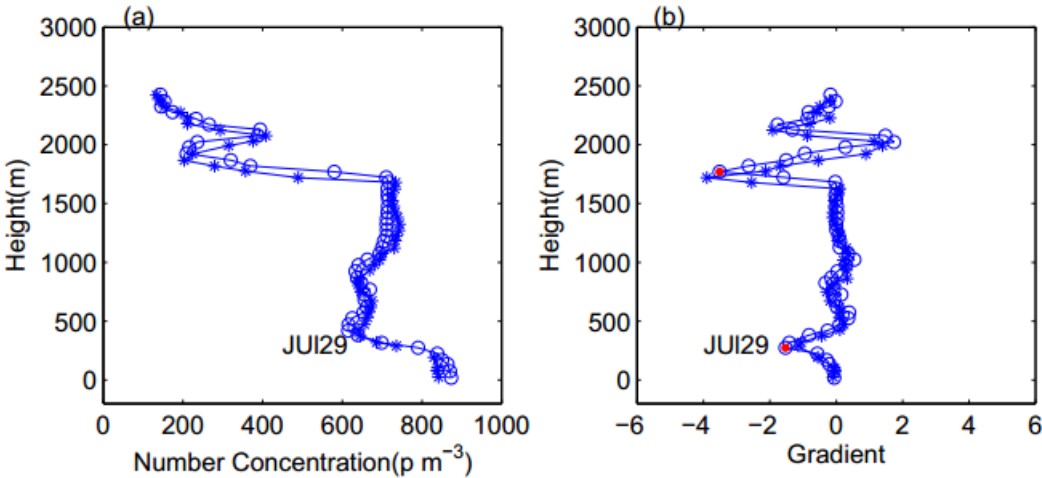

**Figure 5.** Vertical profiles of aerosol number concentration (a) and their gradients (b) from the ascent (open circles) and descent (asterisks) flight in the early morning of July 29 (Flight 4). The red solid circles indicate the tops of mixed layer and residual layer.

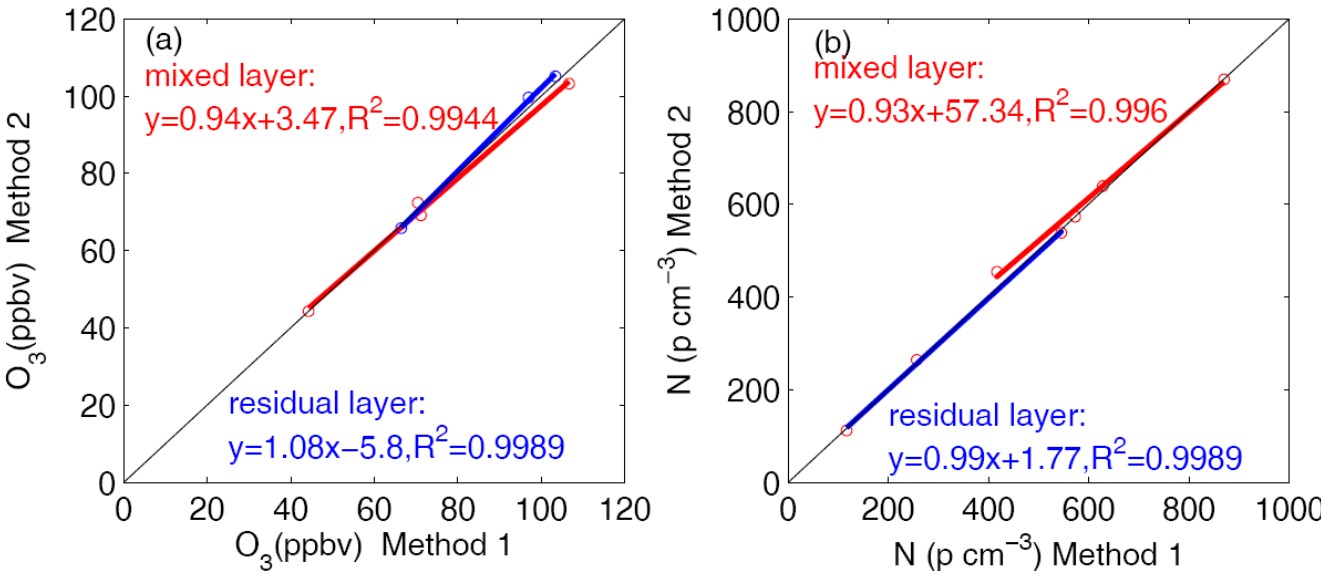

**Figure 6.** Average $O_3$ mixing ratios (a) and aerosol number concentrations (b) in the mixed layer and residual layer obtained using the method of particle extinction gradient (Method 1) in comparison with those obtained using the method of particle number concentration gradient (Method 2).

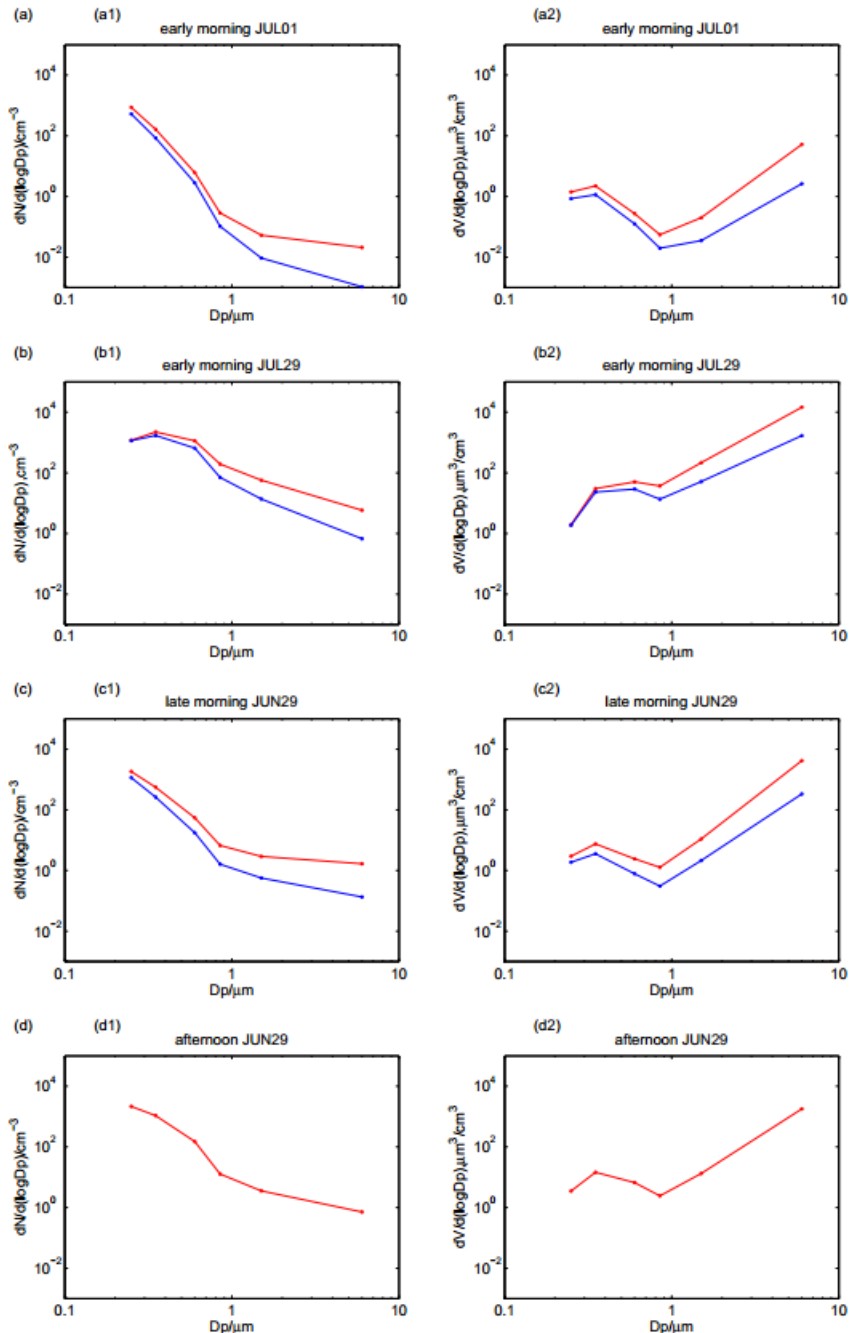

**Figure 7.** Aerosol number and volume size distributions in the mixed layer (red) and residual layer (blue), in the early morning of July 1 (a) and July 29 (b), late morning of June 29 (c) and afternoon of June 29 (d) flights. The mixed layer and residual layer heights are determined using the method of aerosol number concentration gradient.

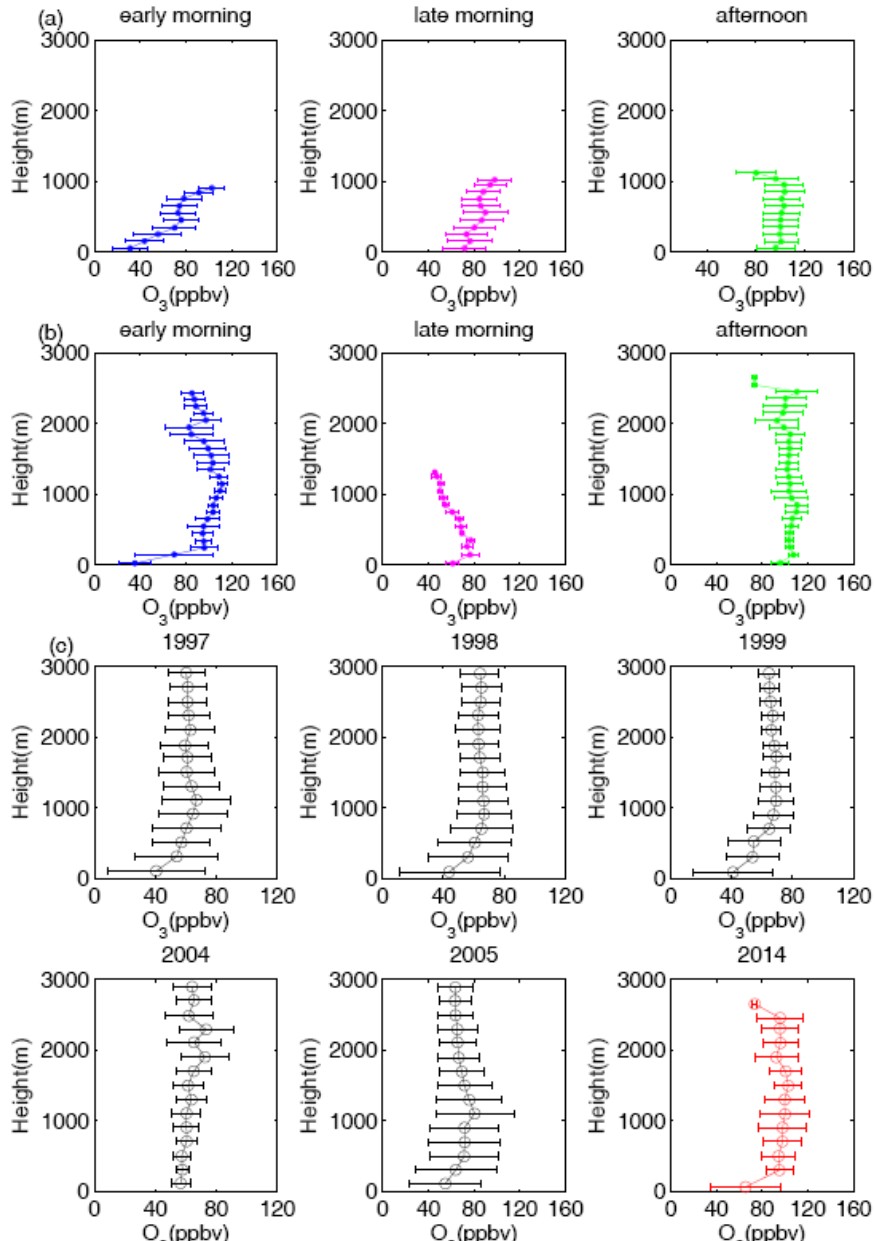

**Figure 8.** Comparison of average $O_3$ vertical profiles observed in the lower troposphere during summer of different years. (a) shows $O_3$ vertical profiles from the tethered balloon experiments during summer of 2014 averaged over early morning (5:00-8:00 LT), late morning (8:00-12:00 LT) and afternoon (12:00-19:00 LT). (b) presents $O_3$ vertical profiles from the UAV experiments during summer of 2014 averaged over early morning (Flights 3, 4 and 5), late morning (Flight 1) and afternoon (Flights 2, 6 and 7). (c) shows $O_3$ vertical profiles averaged over daytime (5:00-19:00 LT) MOZAIC measurements in summer of 1997, 1998, 1999, 2004, 2005 and our Raoyang measurement in summer of 2014. The error bars indicate one standard error of the mean.