# Peer review of "Lower tropospheric distributions of O3 and aerosol over Raoyang, a rural site in the North China Plain"

_Atmospheric Chemistry and Physics, 2016_

## Referee Comment (RC1) · Anonymous Referee #1 · 30 Dec 2016

The authors present measurements of vertical profiles of O3 and aerosol by unmanned aerial vehicle, balloon, and LIDAR over a rural site in the North China Plain (NCP) region. The distributions of O3, aerosol number density, and aerosol scattering property in the mixed layer and residual layer are examined. This new vertical profile data is compared against the previous MOZIAC measurements over the Beijing area to assess the increase in the boundary layer O3 over the NCP region. Overall, the observations are valuable and the interpretation is convincing. The manuscript is clearly organized and well written. I would like to recommend that the paper can be accepted for publication after the following specific comments being addressed.

Specific comments:

1. Page 1, Line 17: change "still quite limited" to "still limited".

2. Page 1, Line 30: In-Service. . .

3. Page 2, Line 5: impacts on human health. . .

4. Page 2, Lines 9-10: "actual vertical distribution of O3 is fundamental. . ." is strange. Please rephrase this sentence.

5. Page 2, Line 18: the authors use "Atmospheric boundary layer (ABL)" throughout the manuscript. In the reviewer's opinion, "planetary boundary layer (PBL)" should be more familiar for the community and readers. The authors should consider to replace the "ABL" by "PBL".

6. Page 4, Line 17, "some other reactive gases": please state what species were measured.

7. Section 2: the authors used a set of miniature analyzers including O3 and aerosol number size distribution monitors for the UAV measurements. Did the authors inter-compare these equipment against the more reliable instruments deployed for the ground-based observations? What's the design of the sampling inlet of the UAV to avoid interference? It would be better if the authors could provide such information, maybe in the supporting materials.

8. Page 5, Line 15 and elsewhere: "vertical profile" instead of "profile".

9. Page 6, Section 3.1: this section consists of only one paragraph which just documents the measurement data with little interpretation. This seems to be not enough as a section. The authors may need to consider either strengthen the discussion of data or combine this paragraph with other sections.

10. Page 7, Line 23: after sunrise. . .

11. Page 10, Line 20: Heilongjiang

12. Section 3.5: the authors discussed the increasing trend of O3 concentrations over the NCP region, especially in the northern part. A recent study reported a significant increase of O3 at a mountain site (Mt. Tai) in the central part of the NCP region. Moreover, this study presents another non-surface measurement effort in this region, and is hence relevant to the present study. The authors may consider to compare their results with this previous effort.

Sun, L., Xue, L. K., Wang, T., Gao, J., Ding, A. J., Cooper, O. R., Lin, M. Y., Xu, P. J., Wang, Z., Wang, X. F., Wen, L., Zhu, Y. H., Chen, T. S., Yang, L. X., Wang, Y., Chen, J. M., and Wang, W. X. Significant increase of summertime ozone at Mount.Tai in Central Eastern China, Atmos. Chem. Phys., 16, 10637-10650, 2016.

13. Page 13, Line 11: the enhancement of 20-41.6 ppbv in O3 concentrations from 2004-2014 points to the rate of 2.0-4.1 ppbv/year of O3 increase. It would be helpful if the authors compare this magnitude of O3 increase with other previous results.

14. Table 1: I presume all the time given here is local time. Please specify.

---

## Referee Comment (RC2) · Anonymous Referee #2 · 24 Jan 2017

This manuscript describes the ozone and particulate matter air quality issues of the North China Plain. The authors then describe the development of an unmanned aerial vehicle (UAV) measurement platform and analyze the results from this measurement platform. The authors did a good job in discussing the impact that the evolution of the planetary boundary layer has on ozone and aerosol concentrations. Generally, the authors described their assumptions well used in the analysis. This manuscript is of interest to the general scientific community because it documents a novel measurement platform and novel measurements in an area that has undergone a recent degradation in its ambient air quality.

General Comments

[Figure]

This manuscript could benefit from some organization and focusing on key findings. The manuscript would be much more focused if the evaluation of the differences between the flights and the HYSPLIT modeling could be moved to supplementary material and simply referenced in the manuscript and focus the manuscript on the ozone and aerosols in the mixed and residual layers and the comparison to MOZAIC observations and trends.

Specific Comments

Abstract lines 21-22: What methods were used to determine the mixed and residual layers? Simply stating that potential temperature profiles and aerosol number concentration should suffice.

Page 2 Line 19: The atmospheric boundary layer (ABL) acronym used later in the manuscript should be defined here. Additionally, the introduction of the friction layer as equivalent to the ABL complicates the sentence and the term friction layer is not used elsewhere and should be removed.

Page 2 Line 22: The sentence "The ABL with vigorous turbulence is often called the mixed layer (or mixing layer)" is not needed because it is discussed in more detail and clarity on page 3 line 1.

Page 3 line 8: "deserves attention and studies" is redundant. "deserves attention." Should suffice.

Page 10 line 24: What exactly does "significantly low" mean? Compared to the other profiles? How was significance determined?

Page 13 line 17: While "severer" is technically correct, "more severe" is more commonly used.

Figure 3: The caption does not state what figures (a), (b) or (c) designate. I assume that they are the same as in Figure 1 but this should be explicitly stated.

[Figure]

---

## Author Comment (AC2) · 1 Mar 2017

**Response to Anonymous Referee #2**

This manuscript could benefit from some organization and focusing on key findings. The manuscript would be much more focused if the evaluation of the differences between the flights and the HYSPLIT modeling could be moved to supplementary material and simply referenced in the manuscript and focus the manuscript on the ozone and aerosols in the mixed and residual layers and the comparison to MOZAIC observations and trends.

**Response**: Thank you for your comments and suggestions. We have moved the figures about airflow fields and airmass trajectories to the supplement. The related text has been modified and moved section 3.1 to strengthen the discussion there as suggested by Referee #1.

Specific Comments
1. Abstract lines 21-22: What methods were used to determine the mixed and residual layers? Simply stating that potential temperature profiles and aerosol number concentration should suffice.

**Response**: The depths of the mixed layer and residual layer were determined using two methods. One is the vertical gradient of LIDAR particle extinction (dE/dz), the other is the vertical gradient of aerosol number concentration (dN/dz). Based on individual dE/dz profiles of particle extinction, we were able to determine the tops of the mixed layer and residual layer for the periods with UAV flights by identifying the minima below 2000 m in the dE/dz profiles if there were no obviously unreasonable values in the vertical profiles due to interferences or noises. In order to select the minima that were really resulted from a strong decrease of aerosol burden for a small height change rather than abnormal variation, we compared the minima in the dE/dz profiles during the flights with those before and after the flights. Finally, the MLD and RLD were obtained according to the heights at which the minima were considered to be the most reasonable ones. Similar to the vertical gradient of particle extinction, the vertical gradient of aerosol number concentration were also used for the determination of mixed and residual layer depths. To find out the key inflection points in the vertical profiles, which indicate the tops of the mixed layer and residual layer, we paid attention to the major structures in vertical profiles and gradients of aerosol number concentration during the ascent and descent flights. Due to the slight differences in time and space in the flight, there were some differences in particle number concentrations measured at the same height between the ascent and descent of a flight. Even though, the t-test ($\alpha=0.05$) helped to judge whether or not the results are accordant.

We have changed the sentence to "The depths of the mixed layer and residual layer were determined according the vertical gradients of LIDAR particle extinction and aerosol number concentration."

   2. Page 2 Line 19: The atmospheric boundary layer (ABL) acronym used later in the manuscript should be defined here. Additionally, the introduction of the friction layer as equivalent to the ABL complicates the sentence and the term friction layer is not used elsewhere and should be removed.

**Response**: Because "planetary boundary layer (PBL)" is more familiar for the community and readers, we have replaced the "ABL" with "PBL" as suggested by Referee #1. According to your suggestion, we have removed the term friction layer.

3. Page 2 Line 22: The sentence "The ABL with vigorous turbulence is often called the mixed layer (or mixing layer)" is not needed because it is discussed in more detail and clarity on page 3 line 1.

**Response**: We have removed the sentence "The ABL with vigorous turbulence is often called the mixed layer (or mixing layer)".

4. Page 3 line 8: "deserves attention and studies" is redundant. "deserves attention."Should suffice.

**Response**: We have removed "and studies".

5. Page 10 line 24: What exactly does "significantly low" mean? Compared to the other profiles? How was significance determined?

**Response**: By "significantly low" we meant that the aerosol number concentration in the early morning of July 1 (Flight 3) was much lower than that in the other two flights (Flight 4 and 5) in the early morning of July 29 and July 31. The original expression was not clear enough. Therefore, we have changed the sentence to "In the whole layer, the aerosol number concentration in the early morning of July 1 (Flight 3) was much lower than those in the other two flights (Flight 4 and 5) in the early morning of July 29 and July 31 (Fig. 1(a2).". Note that the modified text has been moved to section 3.1.

6. Page 13 line 17: While "severer" is technically correct, "more severe" is more commonly used.

**Response**: We have replaced "severer" with "more severe".

7. Figure 3: The caption does not state what figures (a), (b) or (c) designate. I assume that they are the same as in Figure 1 but this should be explicitly stated.

**Response**: Figure 3 shows vertical profiles of particle extinction averaged every two hours. We divided the whole day into three periods, (a) 0:00-7:00, (b) 8:00-15:00 and (c)16:00-23:00 in order to clearly show the average PBL diurnal cycle of vertical distribution of particle extinction. We have changed the figure caption to "Vertical profiles of particle extinction averaged every two hours for (a) 0:00-7:00 LT, (b) 8:00-15:00 LT, and (c) 16:00-23:00 LT".

---

## Author Comment (AC1)

**Response to Anonymous Referee #1**

The authors present measurements of vertical profiles of O3 and aerosol by unmanned aerial vehicle, balloon, and LIDAR over a rural site in the North China Plain (NCP) region. The distributions of O3, aerosol number density, and aerosol scattering property in the mixed layer and residual layer are examined. This new vertical profile data is compared against the previous MOZIAC measurements over the Beijing area to assess the increase in the boundary layer O3 over the NCP region. Overall, the observations are valuable and the interpretation is convincing. The manuscript is clearly organized and well written. I would like to recommend that the paper can be accepted for publication after the following specific comments being addressed.

**Response:** Thank you very much for taking time to review our paper and give valuable comments and suggestions. Please find below our point-by-point responses (in blue).

Specific comments:

1. Page 1, Line 17: change "still quite limited" to "still limited".

Response: Yes, we have deleted the word "quite".

2. Page 1, Line 30: In-Service. . .

**Response:** We have corrected the word.

3. Page 2, Line 5: impacts on human health. . .

Response: We have changed "impacts to human health" to "impacts on human health".

4. Page 2, Lines 9-10: "actual vertical distribution of  $O_3$  is fundamental. . ." is strange. Please rephrase this sentence.

**Response:** We have reworded the text as "The vertical distribution of  $O_3$  is influenced by chemical and meteorological processes and varies with time and location (Kleinman et al., 1994; Fast et al., 1996; Lin et al., 2007; Ma et al., 2011). Therefore, direct measurements are needed to acquire the knowledge about the vertical distribution of  $O_3$ , which is important to understanding atmospheric chemistry and  $O_3$  radiative forcing".

5. Page 2, Line 18: the authors use "Atmospheric boundary layer (ABL)"throughout the manuscript. In the reviewer's opinion, "planetary boundary layer (PBL)" should be more familiar for the community and readers. The authors should consider to replace the "ABL" by "PBL".

**Response:** We have replaced "atmospheric boundary layer" with "planetary boundary layer" and "ABL" with "PBL".

6. Page 4, Line 17, "some other reactive gases": please state what species were measured.

**Response:** Other measured reactive gases include  $NO/NO_2/NO_x$ ,  $NO_y$ , HCHO, PAN, SO2, CO, NH3, etc. We have changed "some other reactive gases" to "some other reactive gases ( $NO/NO_2/NO_x$ ,  $NO_y$ , HCHO, PAN, SO2, CO, NH3, etc.)".

7. Section 2: the authors used a set of miniature analyzers including  $O_3$  and aerosol number size distribution monitors for the UAV measurements. Did the authors inter-compare these equipment against the more reliable instruments deployed for the ground-based observations? What's the design of the sampling inlet of the UAV to avoid interference? It would be better if the authors could provide such information, maybe in the supporting materials.

**Response:** A newly delivered personal ozone monitor (POM, 2B Technologies, USA) was used in our UAV measurements of  $O_3$  vertical profiles. The POM was calibrated in the factory just before delivery using a working standard (Model 205), which was calibrated against a transfer standard (TE 49i-PS, Thermo Electron, USA) traceable to the standard reference photometer #0 (SRP 0) at the US National Institute of Standards and Technology (NIST). We did not calibrate the POM again. However, we confirmed its proper working at the site using an  $O_3$  calibrator (TE 49i-PS, Thermo Electron, USA).

The optical particle counters (OPC) used in our UAV flights were calibrated by the manufacturer Lighthouse following the procedures in accordance with key standards for particle counters including ISO 21501 (https://www.golighthouse.com/en/calibration-and-repairs). The calibration is NIST-traceable. We did not compare the OPC with other instruments at the site.

The sampling inlet that we used in the UAV measurements of aerosol is a stainless steel tube (300 mm, ID 5 mm) with a bullet-shaped head, which has a taper gas way (Figure R1). The sampling inlet was fixed at the front of the UAV nose, parallel to the inlet for airspeed measurement. The tip of the head has an ID of 2.1 mm. The OPC works at a flow rate of 2.83 l/min. These conditions cause an air velocity of about 14 m/s at the tip of the inlet head, which is close to the cruising speed of the UAV (22 m/s). This design of sampling inlet made sure that the aerosol number concentrations were measured under a nearly isokinetic condition. Since a battery-powered UAV was used, there was no interference from engine.

---

## Author Comment (AC3)

**Author response**

Dear editor,
Thank you so much for handling our paper. We are grateful to both referees for their comments and suggestions. We have submitted our point-by-point responses and improved our manuscript according to the referee's suggestions. Below are our responses to referees and revised manuscript with a supplement.
Sincerely,

Xiaobin Xu

**Response to Anonymous Referee #1**

The authors present measurements of vertical profiles of O3 and aerosol by unmanned aerial vehicle, balloon, and LIDAR over a rural site in the North China Plain (NCP) region. The distributions of O3, aerosol number density, and aerosol scattering property in the mixed layer and residual layer are examined. This new vertical profile data is compared against the previous MOZIAC measurements over the Beijing area to assess the increase in the boundary layer O3 over the NCP region. Overall, the observations are valuable and the interpretation is convincing. The manuscript is clearly organized and well written. I would like to recommend that the paper can be accepted for publication after the following specific comments being addressed.

**Response:** Thank you very much for taking time to review our paper and give valuable comments and suggestions. Please find below our point-by-point responses (in blue).

Specific comments:
**1.** Page 1, Line 17: change "still quite limited" to "still limited".

**Response:** Yes, we have deleted the word "quite".

2. Page 1, Line 30: In-Service. . .

**Response:** We have corrected the word.

3. Page 2, Line 5: impacts on human health. . .

**Response:** We have changed "impacts to human health" to "impacts on human health".

4. Page 2, Lines 9-10: "actual vertical distribution of $O_3$ is fundamental. . ." is strange.
Please rephrase this sentence.

**Response:** We have reworded the text as "The vertical distribution of $O_3$ is influenced by chemical and meteorological processes and varies with time and location (Kleinman et al., 1994; Fast et al., 1996; Lin et al., 2007; Ma et al., 2011). Therefore, direct measurements are needed to

acquire the knowledge about the vertical distribution of $O_3$, which is important to understanding atmospheric chemistry and $O_3$ radiative forcing".

5. Page 2, Line 18: the authors use "Atmospheric boundary layer (ABL)"throughout the manuscript. In the reviewer's opinion, "planetary boundary layer (PBL)" should be more familiar for the community and readers. The authors should consider to replace the "ABL" by "PBL".

**Response:** We have replaced "atmospheric boundary layer" with "planetary boundary layer" and "ABL" with "PBL".

6. Page 4, Line 17, "some other reactive gases": please state what species were measured.

**Response:** Other measured reactive gases include $NO/NO_2/NO_x$, $NO_y$, HCHO, PAN, $SO_2$, CO, $NH_3$, etc. We have changed "some other reactive gases" to "some other reactive gases ($NO/NO_2/NO_x$, $NO_y$, HCHO, PAN, $SO_2$, CO, $NH_3$, etc.)".

7. Section 2: the authors used a set of miniature analyzers including $O_3$ and aerosol number size distribution monitors for the UAV measurements. Did the authors inter-compare these equipment against the more reliable instruments deployed for the ground-based observations? What's the design of the sampling inlet of the UAV to avoid interference? It would be better if the authors could provide such information, maybe in the supporting materials.

**Response:** A newly delivered personal ozone monitor (POM, 2B Technologies, USA) was used in our UAV measurements of $O_3$ vertical profiles. The POM was calibrated in the factory just before delivery using a working standard (Model 205), which was calibrated against a transfer standard (TE 49i-PS, Thermo Electron, USA) traceable to the standard reference photometer #0 (SRP 0) at the US National Institute of Standards and Technology (NIST). We did not calibrate the POM again. However, we confirmed its proper working at the site using an $O_3$ calibrator (TE 49i-PS, Thermo Electron, USA).

The optical particle counters (OPC) used in our UAV flights were calibrated by the manufacturer Lighthouse following the procedures in accordance with key standards for particle counters including ISO 21501 (https://www.golighthouse.com/en/calibration-and-repairs). The calibration is NIST-traceable. We did not compare the OPC with other instruments at the site.

The sampling inlet that we used in the UAV measurements of aerosol is a stainless steel tube (300 mm, ID 5 mm) with a bullet-shaped head, which has a taper gas way (Figure R1). The sampling inlet was fixed at the front of the UAV nose, parallel to the inlet for airspeed measurement. The tip of the head has an ID of 2.1 mm. The OPC works at a flow rate of 2.83 l/min. These conditions cause an air velocity of about 14 m/s at the tip of the inlet head, which is close to the cruising speed of the UAV (22 m/s). This design of sampling inlet made sure that the aerosol number concentrations were measured under a nearly isokinetic condition. Since a battery-powered UAV was used, there was no interference from engine.

[Figure]

Figure R1. Schematic of the head of the sampling inlet used in the UAV measurements of aerosol.

We have included the above details in the supplement (section S1) and added "More details are given in the supplement (section S1)." at the end of the 3rd paragraph of section 2.2.

8.  Page 5, Line 15 and elsewhere: "vertical profile" instead of "profile".

**Response:** We use now "vertical profile" instead of "profile" in the whole paper.

9.  Page 6, Section 3.1: this section consists of only one paragraph which just documents the measurement data with little interpretation. This seems to be not enough as a section. The authors may need to consider either strengthen the discussion of data or combine this paragraph with other sections.

**Response:** Indeed, the original section 3.1 lacks some scientific materials. However, we think it is better to give an overview of our vertical profile measurements in the first section in "Results and discussion". We have decided to add some analysis and discussion to this section instead of combining this with the next one. Some vertical profiles show special features (Figs. 1(b1), 1(b2) and 1(a2)), which we discussed in section 3.3 in the previous version. Since Referee #2 suggests to move some of the materials in section 3.3 to the supplement, we have decided to move a part of the text to this section. The following text is added to section 3.1:

"**It is noteworthy in Fig. 1 that aerosol number concentrations during late morning of June 29 (Flights 1) and early morning of July 1 (Flight 3) and the O$_3$ mixing ratio during late morning of June 29 (Flight 1) were significantly lower than those during other flights, and their vertical profiles were slightly different from others. This indicates that some factors might have impacted the levels and vertical profiles of O$_3$ and aerosol. To understand those phenomena, we display the airflow fields at 1000 hPa and 850 hPa over the area surrounding RMS in Fig. S2 in the supplement and 48-h backward trajectories of air parcels arriving at 100 m, 500 m, 1000 m, 1500 m and 2000m over RMS in Fig. S3 in the supplement for 8:00 local time of June 29, July 1, July 29 and July 31, 2014, calculated using the Hybrid Single-Particle Lagrangian Integrated Trajectory (HYSPLIT) model (Draxler and Rolph,**

**2003).**

**Figures S2(a) and S2(b) indicate that the 1000 hPa and 850 hPa levels on early morning of June 29 were dominated by different air circulations. Figure S3(a) shows that the air parcels arriving at 100 m and 500 m over RMS were from the boundary layer over the nearby areas and mainly from south of the site, while those arriving at 1000 m and above originated far from the north (northern Heilongjiang) and travelled rapidly over 2000 m for most of the time. The disparate airflows in the bottom and upper layers made a large negative gradient in the vertical distributions of $O_3$ and aerosol number concentration during Flight 1, as can be seen in the vertical profiles shown in Figs. 1(b1) and 1(b2).**

**In the early morning of July 1 (Flight 3), the aerosol number concentration was significantly low in the whole layer. Figures S2(c) and S2(d) in the supplement show that the airflow at the 1000 hPa and 850 hPa levels over RMS was mostly from the south-southwest. Figure S3(b) in the supplement indicates that air parcels arriving at different heights over RMS originated either from south or from west or east bending to south. Using the Meteorological Information Comprehensive Analysis and Process System (MICAPS, http://www.cma.gov.cn/en2014/20150311/20160615/index.html) we found that there was rainfall within the 48 hours prior to 5:00 LT of July 1 over the north, south by southwest and the southeast of RMS. It can be inferred that in the early morning of July 1 air transported to the lower troposphere over RMS had been mixed with cleaner air during the rainy conditions so that the aerosol number concentration declined substantially. Therefore, synoptic situations are important factors influencing the concentrations of air pollutants and their vertical distributions. Another important factor is the PBL evolution, as discussed in the next section."**

10. Page 7, Line 23: after sunrise. . .

**Response:** We have changed "after the sunrise" to "after sunrise".

11. Page 10, Line 20: Heilongjiang

**Response:** Corrected.

12.    Section 3.5: the authors discussed the increasing trend of $O_3$ concentrations over the NCP region, especially in the northern part. A recent study reported a significant increase of $O_3$ at a mountain site (Mt. Tai) in the central part of the NCP region. Moreover, this study presents another non-surface measurement effort in this region, and is hence relevant to the present study. The authors may consider to compare their results with this previous effort. Sun, L., Xue, L. K., Wang, T., Gao, J., Ding, A. J., Cooper, O. R., Lin, M. Y., Xu, P. J.,Wang, Z., Wang, X. F., Wen, L., Zhu, Y. H., Chen, T. S., Yang, L. X., Wang, Y., Chen, J.M., and Wang, W. X. Significant increase of summertime ozone at Mount.Tai in Central Eastern China, Atmos. Chem. Phys., 16, 10637-10650, 2016.

**Response:** We have compared our results with the relevant results reported in other papers including Sun et al. (2016). Changes have been made in the last two paragraphs of this section.

Please see details in our response to comment #13.

13. Page 13, Line 11: the enhancement of 20-41.6ppbv in $O_3$ concentrations from 2004-2014 points to the rate of 2.0-4.1 ppbv/year of $O_3$ increase. It would be helpful if the authors compare this magnitude of $O_3$ increase with other previous results.

**Response:** Thank you for your suggestion. So far, only a few papers reported rates of change in O3 at sites in the NCP. To compare our results with those from different papers, we have rewritten the fourth paragraph and revised the fifth paragraph in section 3.5 as follows:

"**Figure 8(c) shows that the $O_3$ level over the NCP had experienced a strong positive increase, indicating strengthening photochemical pollution in about two decades. The average mixing ratio of $O_3$ near the ground level had a relatively small increase (8.9 ppb) during 2004-2014, corresponding to an increase rate of about 0.9 ppb/yr. This increase rate is close to the average increase rate (1.1 ppb/yr ) reported by Tang et al. (2009) of surface $O_3$ at six urban/suburban sites in Beijing in July-September during 2001-2006, but only about one third of what Zhang et al. (2014) found for August daytime surface $O_3$ at an urban site in Beijing during 2005-2011 (2.6 ppb/yr). However, the high-end value of $O_3$ near the ground level had a larger increase, as suggested by the right end of the error bars (Fig. 8(c)). This larger increase in the high-end value of $O_3$ is consistent with the large increase of the maximum daily average 8 h (MDA8) mixing ratio of $O_3$ at the Shangdianzi background station (Ma et al., 2016). Compared with the increase of $O_3$ near the ground-level, much larger increases were found in $O_3$ at higher altitudes in the lower troposphere over Raoyang during 2004-2014, with the maximum increase (41.6$\pm$15.5 ppb) being found at 1.5 km. Assuming that the $O_3$ level increased linearly over the ten years, the increase rate would be about 4.2$\pm$1.6 ppb/yr. Sun et al. (2016) compiled and analyzed the $O_3$, and $NO_x$ and CO data collected at the Mt. Tai site (36.25°N, 117.10°E; 1534m asl) during a few campaigns from 2003 to 2015. They reported that $O_3$ at Mt. Tai increased at 1.7$\pm$1.0 ppb/yr in June and 2.1$\pm$0.9 ppb/yr in July-August during 2003-2015. These rates of increase in summer $O_3$ at Mt. Tai and that we obtained for 1.5 km over Raoyang agree within the uncertainties though both sites are about 240 km apart. Based on the MOZAIC measurements over Beijing in summer afternoons (at 15:00–16:00 LT in MJJ) during 1995-2005, Ding et al. (2008) reported an increase rate of about 3 ppb/yr for $O_3$ in 0-2 km. From the data shown in Fig. 8(c), we can obtain an increase rate of 3.3 ppb/yr for summer $O_3$ in 0-2 km over Raoyang for the period 2004-2014, which agrees well with that reported by Ding et al. (2008). Note that the average $O_3$ vertical profile for summer 2014 (Fig. 8(c)) contains measurements from the morning flights so that our estimated increase rate may be significantly lower than that for summer afternoon.**"

"The above comparisons confirm that the abundance of $O_3$ **in the lower troposphere** over the north part of the NCP has largely increased since about two decades. The increase of the $O_3$ level **in summer afternoon period** seems to speed up after 2004. Network observations indicate that surface $O_3$ pollution in China's polluted regions, including the NCP, has become **more severe** in recent years in contrast with the apparent decreases of $PM_{2.5}$ and primary gaseous pollutants (http://www.cnemc.cn/publish/totalWebSite/0492/newList_1.html). Such trend in surface $O_3$ may

exert significant impacts on human health and vegetation. The increase of $O_3$ in the lower troposphere may influence atmospheric chemistry, i.e., increase the oxidation capacity (Ma et al., 2012), and add radiation forcing over the region.*"*

14.Table 1: I presume all the time given here is local time. Please specify.

**Response:** We specify that the time given in Table 1 is local time (LT), and point out that the period in the whole paper is local time.

[revised manuscript text omitted]

**Figure 9̶7.** Aerosol number and volume size distributions in the mixed layer (red) and residual layer (blue), in the early morning of July 1 (a) and July 29 (b), late morning of June 29 (c) and afternoon of June 29 (d) flights. The mixed layer and residual layer heights are determined using the method of aerosol number concentration gradient.

[Figure]

**Figure 8.** Comparison of average $O_3$ vertical profiles observed in the lower troposphere during summer of different years. (a) shows $O_3$ vertical profiles from the tethered balloon experiments during summer of 2014 averaged over early morning (5:00-8:00 LT), late morning (8:00-12:00 LT) and afternoon (12:00-19:00 LT). (b) presents $O_3$ vertical profiles from the UAV experiments during summer of 2014 averaged over early morning (Flights 3, 4 and 5), late morning (Flight 1) and afternoon (Flights 2, 6 and 7). (c) shows $O_3$ vertical profiles averaged over daytime (5:00-19:00 LT) MOZAIC measurements in summer of 1997, 1998, 1999, 2004, 2005 and our Raoyang measurement in summer of 2014. The error bars indicate one standard error of the mean.

*Supplement of*

**Lower tropospheric distributions of O₃ and aerosol over Raoyang, a rural site in the North China Plain**

Rui Wang et al.

*Correspondence to*: Xiaobin Xu (xuxb@camscma.cn)

**S1 Technical details of the UAV measurements**

A newly delivered personal ozone monitor (POM, 2B Technologies, USA) was used in our UAV measurements of $O_3$ vertical profiles. The POM was calibrated in the factory just before delivery using a working standard (Model 205), which was calibrated against a transfer standard (TE 49i-PS, Thermo Electron, USA) traceable to the standard reference photometer #0 (SRP 0) at the US National Institute of Standards and Technology (NIST). We did not calibrate the POM again. However, we confirmed its proper working at the site using an $O_3$ calibrator (TE 49i-PS, Thermo Electron, USA).

The optical particle counters (OPC) used in our UAV flights were calibrated by the manufacturer Lighthouse following the procedures in accordance with key standards for particle counters including ISO 21501 (https://www.golighthouse.com/en/calibration-and-repairs). The calibration is NIST-traceable. We did not compare the OPC with other instruments at the site.

The sampling inlet that we used in the UAV measurements of aerosol is a stainless steel tube (300 mm, ID 5 mm) with a bullet-shaped head, which has a taper gas way (Figure S1). The sampling inlet was fixed at the front of the UAV nose, parallel to the inlet for airspeed measurement. The tip of the head has an ID of 2.1 mm. The OPC works at a flow rate of 2.83 l/min. These conditions cause an air velocity of about 14 m/s at the tip of the inlet head, which is close to the cruising speed of the UAV (22 m/s). This design of sampling inlet made sure that the aerosol number concentrations were measured under a nearly isokinetic condition. Since a battery-powered UAV was used, there was no interference from engine.

[Figure]

**Figure S1.** Schematic of the head of the sampling inlet used in the UAV measurements of aerosol.

**S2 Airflow fields and air-mass backward trajectories**

[Figure]

**Figure S2.** Geopotential heights (see color scale) and streamlines over the area surrounding Raoyang at 8:00 of June 29, 2014 at 850 hPa (a) and 1000 hPa (b); at 8:00 of July 1, 2014 at 850 hPa (c) and 1000 hPa (d); at 8:00 of July 29, 2014 at 850 hPa (e) and 1000 hPa (f); at 8:00 of July 31, 2014 at 850 hPa (g) and 1000 hPa (h).The reanalysis data from ECWMF were used to make these plots.

[Figure]

**Figure S3.** 48-h backward trajectories ending at 100 m, 500 m, 1000 m, 1500 m and 2000 m over Raoyang at 8:00 of June 29 (a), July 1 (b), July 29 (c) and July31 (d), 2014.

**S3 Average diurnal variations of O₃ and NO**

[Figure]

**Figure S4.** Average diurnal variations of surface O$_3$ and NO at Raoyang during the campaign in June-August 2014. The error bars represent one standard error of the mean.